# Temporal Trend of Multimorbidity of Noncommunicable Diseases among Brazilian Adults, 2006–2021

**Thaís C. M. Caldeira** [1,*] **, Taciana M. Sousa** [2] **, Marcela M. Soares** [1] **, Izabella P. A. Veiga** [2] **, Luiza E. S. Silva** [1] **and Rafael M. Claro** [2]

1   Department of Preventive and Social Medicine, Universidade Federal de Minas Gerais, Belo Horizonte 30130-100, Brazil

2   Nutrition Department, Universidade Federal de Minas Gerais, Belo Horizonte 30130-100, Brazil; rafael.claro@gmail.com (R.M.C.)

*   Correspondence: thaismarquezine@ufmg.com; Tel.: +55-31-99129-8297

**Abstract:** We aimed to identify the temporal trend of multimorbidity of noncommunicable disease (NCDs) among Brazilian adults (n = 784,479) over a 16-year period of time. This is a time series of cross-sectional studies based on data from the Surveillance System of Risk and Protective Factors for Chronic Diseases by Telephone Survey (Vigitel) from 2006 to 2021. The presence of multimorbidity was assessed from the co-occurrence of (1) obesity and diabetes; (2) obesity and hypertension; and (3) obesity and diabetes and/or hypertension. Linear regression models (Prais–Winsten) were used to identify significant trends for the complete period (2006–2021) and the most recent quinquennium (2017–2021). Multimorbidity of obesity and diabetes and/or hypertension increased in the complete period (5.5% to 9.6%; 0.22 pp/year) and the most recent period (8.3% to 9.6%; 0.40 pp/year) studied. The highest increase occurred especially among men, older adults, and those with fewer years of education. Additionally, there was a high prevalence and an intense increase in multimorbidity among adults with poor self-rated health. These results reinforce the need for expanding and strengthening public health actions focused on individuals with multimorbidity especially with obesity.

**Keywords:** multimorbidity; obesity; public health; chronic disease; Brazil



## 1. Introduction

Noncommunicable diseases (NCDs) have increased considerably and are a current major public health problem [1]. Obesity is rising worldwide [2], and is one of the leading causes of death globally, accounting for around 8.8% of all deaths in 2019 [3]. In addition to being a disease, obesity is also a risk factor for other NCDs, such as diabetes [4], hypertension, and heart diseases, among others [5], contributing to an increased chance of developing these diseases and worsening their prognosis [4,5].

In recent years, there has been an increasing interest in multimorbidity, commonly defined as the co-occurrence of at least two chronic conditions in the same individual [6]. This interest has been prompted in particular by the great challenge for public health caused by the co-occurrence of diseases. Recent studies have shown that multimorbidity occurs frequently in the world population due to changes in lifestyle factors, notably physical inactivity and obesity [7], along with population ageing [8,9]. Individuals with multimorbidity are more likely to use health services, such as primary care, or even to be admitted to hospital, and exhibit high rates of readmission [10,11]. This implies considerably high financial expenditures related to disease management [12]. Furthermore, multimorbidity increases the chances of premature death [13].

Previous studies involving the Brazilian population have shown an increased prevalence of obesity and the isolated presence of NCDs over the years. Additionally, they have predicted its increase in the coming years [14–16]. However, these studies were mainly

focused on analyzing the isolated presence of these diseases among individuals, in response to the need to update the temporal trend of the presence of these multimorbidities. Thus, a better understanding of multimorbidity trends is necessary to support public policies within the Brazilian Unified Health System (SUS, *Sistema Único de Saúde*, in Portuguese).

Population health surveys make it possible to monitor the prevalence of NCDs. As a result, since 2006, Brazil has relied on the Surveillance System of Risk and Protective Factors for Chronic Diseases by Telephone Survey to carry out this monitoring and to support the development of public policy [17]. Thus, the present study aimed to identify the temporal trend of multimorbidity of NCDs among Brazilian adults over a 16-year time period.

## 2. Methods

A time series analysis of cross-sectional surveys, based on data collected by the Surveillance System of Risk and Protective Factors for Chronic Diseases by Telephone Survey (Vigitel) from 2006 to 2021, was performed. Vigitel is an annual survey based on interviews conducted through landline phone calls with adult individuals ($\geq$18 years) living in the 26 Brazilian state capitals and the Federal District [17].

Vigitel's sampling process is divided into two stages. Firstly, 10,000 landlines in each city are randomly selected from the phone registers provided by the main national telephone companies. These lines are organized into replicas of 200 landlines (this is done by reproducing the same proportion of lines as the original registration). The second stage consists of identifying eligible landlines to conduct the interview and randomly selecting one of the adults living in the house to answer the questions in the interview [17]. A minimum sample size of approximately 2000 interviews per year in each city was established in order to estimate the frequency of all indicators with a maximum error of two percentage points and a 95% confidence interval (95% CI) [17]. Samples from approximately 1000 individuals were accepted from the year 2020 onwards due to the COVID-19 pandemic [17].

The Vigitel estimates are weighted to represent the total adult population of each city. The final weight consists of two factors, the first of which deals with the likelihood of unequal sampling occurring of households with more than one landline and more than one resident, whereas the second factor compares the distribution of the interviewed population to that projected for the entire population in each study site and year (using the Rake method), based on official projections for the population [17]. Further details regarding the sampling process used in Vigitel can be found in the system's annual reports [17].

Questions regarding the self-reported medical diagnosis of diabetes and hypertension, as well of the presence of obesity, were used in the present investigation. The presence of obesity was determined based on self-reported information about current weight and height. Vigitel collected this information with the questions: "Do you know your weight (even if it is an approximate value)?" and "Do you know your height?" [17]. Body Mass Index (BMI) was obtained by dividing the weight, in kilograms, by the square of the height, in meters. The cutoff points recommended by the World Health Organization (WHO) were used to identify obesity (BMI $\geq$ 30 kg/m$^2$) [18]. The presence of diabetes and hypertension was established based on an affirmative response to the question "Has a doctor ever told you that you have [diabetes or high pressure]?" (Yes | No). The presence of multimorbidity was assessed from of simultaneous occurrence of (1) obesity and diabetes; (2) obesity and hypertension; and (3) obesity and diabetes and/or hypertension.

A set of sociodemographic characteristics, such as sex (men and women), age group (18 to 24, 25 to 34, 35 to 44, 45 to 54, 55 to 64 and $\geq$65 years old), schooling years (0 to 8, 9 to 11 and $\geq$12 years) and poor self-rated health (adults who self-rated their health as poor or very poor) complemented the analyses.

*Statistical Analysis*

The prevalence of obesity and three multimorbidity combinations were obtained for each year for the complete set of population and sociodemographic groups. Linear regression models (Prais–Winsten) were used to identify significant trends (increase or decrease) in the prevalence of multimorbidity in the analyzed period. Coefficients from these models indicated changes in percentage points per year (pp/year) in the prevalence of multimorbidity during the analyzed period. Significant values observed for this coefficient (*p*-value < 0.05) indicated consistent and significant variations in the time series. This analysis was applied to the complete period (2006–2021) and to the most recent period (2017–2021).

The data were organized and analyzed in the Stata software version 16.1 Vigitel's databases are available for public use at the official Brazilian Ministry of Health (MoH) website (<http://svs.aids.gov.br/download/Vigitel/> accessed on 21th February 2023). Vigitel is approved by the Brazilian Committee on Ethics in Research with Human Beings of MoH (65610017.1.0000.0008).

## 3. Results

From 2006 to 2021, 784,479 adults (≥18 years) were interviewed by Vigitel. Obesity prevalence increased from 11.8% to 22.4% between 2006 and 2021 (0.66 pp/year), with a significant increase for all sociodemographic groups. Regarding sex, the highest increase was among women (0.67 pp/year). For age groups, the greatest increases were among adults aged 25 to 54 years (0.69 pp/year between 25 and 34 years; 0.81 pp/year between 35 and 44 years and 0.57 pp/year between 45 and 54 years). For schooling levels, the greatest increase was among adults with 9 to 11 years of study (0.83 pp/year). Among those with poor health, obesity increased from 23.9% to 45.2% (0.98 pp/year). In the most recent period (2017–2021), an increase in obesity was also observed for almost all sociodemographic groups, especially among women (0.97 pp/year), individuals aged 45 to 54 years (1.06 pp/year) and adults with 12 or more years of study (1.08 pp/year) (Table 1).

**Table 1.** Prevalence (%) of obesity among Brazilian adults (≥18 years), by year, according to sociodemographic characteristics and self-rated health. Vigitel Brazil, 2006–2021.

| Variables | 2006 | 2007 | 2008 | 2009 | 2010 | 2011 | 2012 | 2013 | 2014 | 2015 | 2016 | 2017 | 2018 | 2019 | 2020 | 2021 | Annual Variation pp/year (2006/21) | Annual Variation pp/year (2017/21) |
|---|---|---|---|---|---|---|---|---|---|---|---|---|---|---|---|---|---|---|
| **Sex** | | | | | | | | | | | | | | | | | | |
| Men | 11.4 | 13.6 | 13.4 | 13.9 | 14.4 | 15.5 | 16.5 | 17.5 | 17.6 | 18.1 | 18.1 | 19.2 | 18.7 | 19.5 | 20.3 | 22.0 | 0.61 ** | 0.73 * |
| Women | 12.1 | 13.1 | 13.9 | 14.7 | 15.6 | 16.5 | 18.2 | 17.5 | 18.2 | 19.7 | 19.6 | 18.7 | 20.7 | 21.0 | 22.6 | 22.6 | 0.67 ** | 0.97 ** |
| **Age group (years)** | | | | | | | | | | | | | | | | | | |
| 18 to 24 | 4.4 | 4.1 | 4.8 | 6.5 | 5.7 | 5.7 | 7.5 | 6.3 | 8.5 | 8.3 | 8.5 | 9.2 | 7.4 | 8.7 | 9.9 | 12.2 | 0.42 ** | 0.89 |
| 25 to 34 | 9.8 | 11.4 | 11.2 | 11.9 | 12.2 | 13.7 | 15.1 | 15.0 | 15.1 | 17.9 | 17.1 | 16.5 | 18.0 | 19.3 | 19.6 | 20.8 | 0.69 ** | 1.00 * |
| 35 to 44 | 12.8 | 14.9 | 15.2 | 15.6 | 16.6 | 19.6 | 19.7 | 20.1 | 22.0 | 23.6 | 22.5 | 22.3 | 23.2 | 22.8 | 24.7 | 25.5 | 0.81 ** | 0.79 * |
| 45 to 54 | 16.1 | 19.5 | 18.6 | 17.9 | 21.6 | 21.2 | 22.6 | 22.5 | 21.3 | 21.7 | 22.8 | 23.3 | 24.0 | 24.5 | 27.1 | 26.2 | 0.57 ** | 1.06 * |
| 55 to 64 | 18.0 | 19.5 | 20.8 | 21.6 | 19.8 | 21.1 | 23.4 | 24.4 | 23.1 | 22.7 | 22.9 | 22.6 | 24.6 | 24.3 | 26.2 | 26.2 | 0.46 ** | 0.86 ** |
| ≥65 years | 16.1 | 15.6 | 17.4 | 17.8 | 19.4 | 17.7 | 19.0 | 20.2 | 19.8 | 19.4 | 20.3 | 20.3 | 21.5 | 20.9 | 20.2 | 21.8 | 0.35 ** | 0.00 |
| **Schooling (years)** | | | | | | | | | | | | | | | | | | |
| 0 to 8 | 15.3 | 16.9 | 17.5 | 18.1 | 18.8 | 19.7 | 21.7 | 22.3 | 22.7 | 23.6 | 23.5 | 23.3 | 24.5 | 24.2 | 25.3 | 25.8 | 0.69 ** | 0.55 * |
| 9 to 11 | 9.0 | 10.7 | 11.0 | 12.2 | 13.1 | 14.2 | 15.2 | 15.1 | 17.2 | 17.8 | 18.3 | 17.8 | 19.4 | 19.9 | 20.8 | 22.8 | 0.83 ** | 1.06 * |
| ≥12 years | 8.6 | 9.9 | 10.2 | 10.6 | 11.7 | 13.0 | 14.4 | 14.3 | 12.3 | 14.6 | 14.9 | 16.0 | 15.8 | 17.2 | 19.3 | 19.0 | 0.65 ** | 1.08 * |
| **Poor self-rated health** | 23.9 | 26.7 | 26.3 | 28.3 | 26.4 | 29.8 | 34.8 | 31.3 | 29.4 | 31.6 | 35.8 | 34.1 | 37.3 | 38.8 | 33.7 | 45.2 | 0.98 ** | 0.77 |
| **Total** | 11.8 | 13.3 | 13.7 | 14.3 | 15.1 | 16.0 | 17.4 | 17.5 | 17.9 | 18.9 | 18.9 | 18.9 | 19.8 | 20.3 | 21.5 | 22.4 | 0.66 ** | 0.86 * |

\* *p* < 0.05; \*\* *p* < 0.001; n = 784,479.

The prevalence of the co-occurrence of obesity and diabetes increased from 1.5% to 3.1% between 2006 and 2021 (0.10 pp/year), with an increase for all sociodemographic groups. The highest increases were among women (0.11 pp/year) and adults 55 and over (0.15 pp/year between 55 and 64 years and 0.20 pp/year for those aged 65 and over). Significant increases in the co-occurrence of obesity and diabetes were observed among adults with the lowest schooling (0.24 pp/year). Poor self-rated health among those individuals with obesity and diabetes increased from 5.7% to 11.1% (0.26 pp/year). For the most recent period (2017–2021), the greatest increases were among women (0.18 pp/year), individuals aged 65 and over (0.22 pp/year), and in adults with 9 to 11 years of schooling (0.21 pp/year) and with a greater increase in poor self-rated health (0.65 pp/year) (Table 2).

**Table 2.** Prevalence (%) of co-occurrence of obesity and diabetes among Brazilian adults (≥18 years), by year, according to sociodemographic characteristics and self-rated health. Vigitel Brazil, 2006–2021.

| Variables | 2006 | 2007 | 2008 | 2009 | 2010 | 2011 | 2012 | 2013 | 2014 | 2015 | 2016 | 2017 | 2018 | 2019 | 2020 | 2021 | Annual Variation pp/year (2006/21) | Annual Variation pp/year (2017/21) |
|---|---|---|---|---|---|---|---|---|---|---|---|---|---|---|---|---|---|---|
| **Sex** | | | | | | | | | | | | | | | | | | |
| Men | 1.0 | 1.1 | 1.4 | 1.5 | 1.5 | 1.3 | 1.9 | 1.9 | 2.0 | 1.8 | 2.3 | 2.1 | 2.2 | 2.3 | 2.3 | 2.5 | 0.09 ** | 0.11 * |
| Women | 1.9 | 1.9 | 2.1 | 1.9 | 2.5 | 2.2 | 2.9 | 2.4 | 2.9 | 2.7 | 3.7 | 2.7 | 3.1 | 2.9 | 3.4 | 3.5 | 0.11 ** | 0.18 * |
| **Age group (years)** | | | | | | | | | | | | | | | | | | |
| 18 to 24 | 0.1 | 0.0 | 0.0 | 0.1 | 0.2 | 0.1 | 0.4 | 0.0 | 0.0 | 0.1 | 0.3 | 0.1 | 0.1 | 0.1 | 0.4 | 0.3 | 0.01 * | 0.09 * |
| 25 to 34 | 0.4 | 0.5 | 0.2 | 0.2 | 0.5 | 0.3 | 0.3 | 0.5 | 0.4 | 0.3 | 0.6 | 0.4 | 0.7 | 0.6 | 0.9 | 0.8 | 0.03 * | 0.12 ** |
| 35 to 44 | 0.9 | 1.0 | 1.4 | 0.9 | 1.3 | 1.1 | 1.2 | 1.8 | 1.5 | 2.1 | 2.2 | 1.4 | 1.4 | 1.6 | 1.6 | 1.8 | 0.06 * | 0.10 * |
| 45 to 54 | 2.3 | 2.3 | 2.9 | 2.3 | 3.4 | 2.8 | 4.1 | 2.9 | 3.6 | 3.1 | 4.2 | 3.5 | 3.9 | 3.4 | 3.3 | 4.4 | 0.10 ** | 0.01 |
| 55 to 64 | 4.6 | 3.9 | 4.4 | 5.2 | 4.3 | 4.8 | 7.0 | 5.1 | 6.6 | 5.3 | 6.5 | 5.8 | 5.8 | 5.9 | 7.1 | 5.9 | 0.15 ** | 0.27 |
| ≥65 years | 4.2 | 4.2 | 5.7 | 5.1 | 5.9 | 4.9 | 5.6 | 5.8 | 6.5 | 5.6 | 8.0 | 6.5 | 6.6 | 7.2 | 6.9 | 7.6 | 0.20 ** | 0.22 * |
| **Schooling (years)** | | | | | | | | | | | | | | | | | | |
| 0 to 8 | 2.6 | 2.2 | 3.1 | 2.8 | 3.2 | 3.0 | 3.9 | 4.0 | 4.4 | 4.0 | 5.7 | 4.9 | 5.2 | 5.4 | 5.5 | 5.7 | 0.24 ** | 0.19 * |
| 9 to 11 | 0.6 | 1.0 | 0.9 | 0.9 | 1.4 | 1.2 | 1.8 | 1.2 | 1.6 | 1.5 | 1.9 | 1.7 | 1.9 | 2.0 | 2.4 | 2.4 | 0.11 ** | 0.21 ** |
| ≥12 years | 0.6 | 0.9 | 0.7 | 1.0 | 1.1 | 0.9 | 1.2 | 1.0 | 1.3 | 1.3 | 1.4 | 1.0 | 1.2 | 1.0 | 1.5 | 1.7 | 0.05 ** | 0.18 * |
| **Poor self-rated health** | 5.7 | 4.9 | 9.0 | 7.1 | 7.7 | 7.8 | 8.8 | 7.7 | 6.6 | 7.0 | 11.6 | 8.6 | 8.0 | 10.1 | 9.4 | 11.1 | 0.26 * | 0.65 * |
| **Total** | 1.5 | 1.5 | 1.8 | 1.7 | 2.1 | 1.8 | 2.4 | 2.2 | 2.5 | 2.3 | 3.0 | 2.4 | 2.7 | 2.7 | 2.9 | 3.1 | 0.10 * | 0.15 * |

Co-occurrence of obesity plus diabetes. * $p < 0.05$; ** $p < 0.001$; n = 784,479.

The prevalence of co-occurrence of obesity and hypertension increased from 5.2% to 8.7% between 2006 and 2021 (0.19 pp/year) for the total population and all sociodemographic groups. Regarding sex, the highest increase was found among men (0.21 pp/year). Regarding age groups, the highest increases were among adults aged 55 and over (0.24 pp/year between 55 and 64 years and 0.22 pp/year for those aged 65 and over). The highest increases in this co-occurrence were among adults with the lowest number of schooling years (0.41 pp/year). Poor self-rated health increased from 15.22% to 23.8% (0.39 pp/year). In the most recent period (2017–2021), an increase in the prevalence of co-occurrence between obesity and hypertension was also observed for most sociodemographic groups. The greatest increases were found among men (0.44 pp/year), adults aged 35 to 54 years (0.32 pp/year for 35 to 44 years and 0.72 pp/year for 45 to 54 years), and adults with lower levels of schooling (0.75 pp/year), in addition to the highest increase being among those with poor self-rated health (0.61 pp/year) (Table 3).

The prevalence of co-occurrence of obesity and diabetes and/or hypertension increased from 5.5% to 9.6% between 2006 and 2021 (0.22 pp/year) for the total population and for all sociodemographic groups. The increment was similar regardless of sex (0.23 pp/year for men and 0.21 pp/year for women) between 2006 and 2021. For age groups, the highest increases were found among adults aged 55 and over (0.29 pp/year for 55 and 64 years and 0.24 pp/year for 65 years and over). The highest increase occurred among adults with less schooling (0.45 pp/year for 0 to 8 years of study and 0.30

pp/year for 9 to 11 years of study). Poor self-rated health increased from 15.6% to 27.3% (0.50 pp/year). For the most recent period (2017–2021), there was an increase in the prevalence of multimorbidity, mainly among men (0.45 pp/year), adults aged 45 to 64 years (0.63 pp/year for 45 to 54 years and 0.50 pp/year for 55 to 64 years) and adults with 0 to 8 years of schooling (0.79 pp/year) (Table 4).

**Table 3.** Prevalence (%) of co-occurrence of obesity and hypertension among Brazilian adults (≥18 years), by year, according to sociodemographic characteristics and self-rated health. Vigitel Brazil, 2006–2021.

| Variables | 2006 | 2007 | 2008 | 2009 | 2010 | 2011 | 2012 | 2013 | 2014 | 2015 | 2016 | 2017 | 2018 | 2019 | 2020 | 2021 | Annual Variation pp/year (2006/21) | Annual Variation pp/year (2017/21) |
|---|---|---|---|---|---|---|---|---|---|---|---|---|---|---|---|---|---|---|
| **Sex** | | | | | | | | | | | | | | | | | | |
| Men | 4.1 | 5.2 | 6.0 | 5.8 | 6.0 | 6.0 | 6.1 | 6.9 | 7.0 | 6.3 | 7.2 | 7.1 | 6.9 | 7.1 | 8.3 | 8.3 | 0.21 ** | 0.44 * |
| Women | 6.2 | 6.5 | 7.6 | 7.6 | 7.7 | 8.0 | 8.8 | 8.4 | 8.2 | 8.8 | 9.0 | 8.3 | 9.3 | 9.0 | 9.5 | 9.1 | 0.18 ** | 0.16 |
| **Age group (years)** | | | | | | | | | | | | | | | | | | |
| 18 to 24 | 0.5 | 0.8 | 0.9 | 0.8 | 1.0 | 0.6 | 0.6 | 0.6 | 1.0 | 0.8 | 1.3 | 1.1 | 1.2 | 1.1 | 0.9 | 0.9 | 0.03 * | −0.07 |
| 25 to 34 | 2.0 | 3.1 | 3.3 | 3.4 | 2.5 | 3.1 | 3.2 | 2.9 | 3.3 | 3.6 | 3.8 | 3.3 | 3.3 | 3.5 | 3.6 | 4.5 | 0.08 * | 0.24 * |
| 35 to 44 | 4.5 | 5.4 | 6.5 | 5.9 | 6.0 | 7.0 | 6.7 | 7.2 | 7.4 | 7.8 | 7.0 | 6.7 | 6.9 | 6.7 | 8.2 | 7.0 | 0.14 * | 0.32 * |
| 45 to 54 | 8.5 | 9.9 | 10.7 | 10.2 | 12.2 | 11.2 | 12.0 | 12.0 | 10.6 | 11.3 | 12.0 | 10.9 | 11.6 | 11.9 | 14.1 | 11.5 | 0.17 * | 0.72 * |
| 55 to 64 | 12.3 | 11.8 | 14.3 | 14.6 | 13.8 | 14.5 | 16.1 | 15.9 | 16.0 | 14.1 | 15.0 | 14.6 | 16.5 | 15.0 | 16.0 | 17.4 | 0.24 * | 0.38 |
| ≥65 years | 12.1 | 11.5 | 13.6 | 13.7 | 13.4 | 13.0 | 13.9 | 14.9 | 14.2 | 13.6 | 16.1 | 15.3 | 14.8 | 15.4 | 15.2 | 15.2 | 0.22 ** | 0.05 |
| **Schooling (years)** | | | | | | | | | | | | | | | | | | |
| 0 to 8 | 7.9 | 8.3 | 10.3 | 10.3 | 10.3 | 10.3 | 11.6 | 12.1 | 11.7 | 12.1 | 13.1 | 12.2 | 13.6 | 13.3 | 15.4 | 14.4 | 0.41 ** | 0.75 ** |
| 9 to 11 | 3.1 | 4.0 | 4.4 | 4.5 | 5.0 | 5.2 | 5.4 | 5.2 | 6.4 | 5.7 | 6.8 | 6.4 | 6.8 | 7.1 | 7.6 | 7.9 | 0.27 ** | 0.38 ** |
| ≥12 years | 2.8 | 3.8 | 3.9 | 3.8 | 3.8 | 4.8 | 4.7 | 4.9 | 4.0 | 4.7 | 4.7 | 5.0 | 4.7 | 4.8 | 5.3 | 5.2 | 0.12 ** | 0.13 |
| **Poor self-rated health** | 15.2 | 16.1 | 19.4 | 17.3 | 16.6 | 20.2 | 20.9 | 18.9 | 16.3 | 19.3 | 22.5 | 21.4 | 19.6 | 20.8 | 20.8 | 23.8 | 0.39 * | 0.61 |
| **Total** | 5.2 | 5.9 | 6.9 | 6.8 | 6.9 | 7.1 | 7.5 | 7.7 | 7.7 | 7.7 | 8.2 | 7.7 | 8.2 | 8.1 | 9.0 | 8.7 | 0.19 ** | 0.32 * |

Co-occurrence of obesity plus arterial hypertension. * $p < 0.05$; ** $p < 0.001$; n = 784,479.

**Table 4.** Prevalence (%) of co-occurrence of obesity and diabetes and/or hypertension among Brazilian adults (≥18 years), by year, according to sociodemographic characteristics and self-rated health. Vigitel Brazil, 2006–2021.

| Variables | 2006 | 2007 | 2008 | 2009 | 2010 | 2011 | 2012 | 2013 | 2014 | 2015 | 2016 | 2017 | 2018 | 2019 | 2020 | 2021 | Annual Variation pp/year (2006/21) | Annual Variation pp/year (2017/21) |
|---|---|---|---|---|---|---|---|---|---|---|---|---|---|---|---|---|---|---|
| **Sex** | | | | | | | | | | | | | | | | | | |
| Men | 4.5 | 5.7 | 6.3 | 6.2 | 6.3 | 6.4 | 6.5 | 7.5 | 7.8 | 6.9 | 7.9 | 7.6 | 7.4 | 7.7 | 8.9 | 9.1 | 0.23 ** | 0.45 * |
| Women | 6.5 | 7.0 | 7.8 | 7.9 | 8.3 | 8.4 | 9.4 | 8.9 | 8.8 | 9.4 | 9.8 | 8.8 | 9.8 | 9.6 | 10.4 | 10.0 | 0.21 ** | 0.30 * |
| **Age group (years)** | | | | | | | | | | | | | | | | | | |
| 18 to 24 | 0.5 | 0.8 | 0.9 | 0.9 | 1.2 | 0.7 | 0.8 | 0.6 | 1.0 | 0.8 | 1.5 | 1.1 | 1.2 | 1.2 | 0.9 | 1.2 | 0.03 * | −0.04 |
| 25 to 34 | 2.1 | 3.3 | 3.4 | 3.5 | 2.7 | 3.3 | 3.4 | 3.3 | 3.5 | 3.7 | 4.2 | 3.5 | 3.5 | 3.6 | 4.1 | 5.1 | 0.10 ** | 0.38 * |
| 35 to 44 | 4.8 | 5.8 | 6.7 | 6.2 | 6.6 | 7.2 | 7.2 | 8.1 | 8.0 | 8.5 | 7.8 | 7.2 | 7.4 | 7.2 | 9.1 | 8.0 | 0.18 * | 0.47 |
| 45 to 54 | 9.0 | 10.5 | 11.1 | 10.8 | 12.8 | 11.8 | 13.0 | 12.4 | 12.0 | 12.3 | 13.1 | 11.9 | 12.5 | 12.8 | 14.5 | 12.7 | 0.20 ** | 0.63 * |
| 55 to 64 | 12.8 | 12.4 | 14.9 | 15.2 | 14.5 | 15.1 | 17.3 | 16.6 | 17.3 | 15.1 | 16.2 | 15.6 | 17.4 | 16.2 | 17.8 | 18.3 | 0.29 ** | 0.50 * |
| ≥65 years | 13.0 | 12.6 | 14.1 | 14.3 | 14.5 | 13.8 | 14.6 | 15.9 | 15.5 | 14.6 | 16.9 | 16.3 | 15.9 | 16.5 | 16.2 | 16.4 | 0.24 ** | 0.08 |
| **Schooling (years)** | | | | | | | | | | | | | | | | | | |
| 0 to 8 | 8.3 | 9.1 | 10.7 | 10.7 | 11.0 | 10.8 | 12.3 | 13.1 | 12.8 | 13.1 | 14.1 | 13.1 | 14.4 | 14.3 | 16.5 | 15.4 | 0.45 ** | 0.79 * |
| 9 to 11 | 3.3 | 4.3 | 4.5 | 4.8 | 5.4 | 5.5 | 6.0 | 5.6 | 6.8 | 6.1 | 7.4 | 6.8 | 7.3 | 7.6 | 8.3 | 8.6 | 0.30 ** | 0.48 ** |
| ≥12 years | 3.0 | 3.9 | 4.1 | 4.1 | 4.1 | 5.0 | 5.1 | 5.2 | 4.5 | 5.1 | 5.4 | 5.3 | 5.0 | 5.1 | 5.9 | 6.0 | 0.15 ** | 0.24 |
| **Poor self-rated health** | 15.6 | 16.9 | 20.1 | 18.3 | 18.4 | 20.6 | 22.2 | 20.0 | 17.2 | 20.5 | 23.3 | 22.4 | 21.0 | 22.4 | 22.5 | 27.3 | 0.50 ** | 1.06 |
| **Total** | 5.5 | 6.4 | 7.1 | 7.1 | 7.4 | 7.4 | 8.1 | 8.2 | 8.4 | 8.3 | 8.9 | 8.3 | 8.7 | 8.7 | 9.7 | 9.6 | 0.22 ** | 0.40 * |

Co-occurrence of obesity and diabetes and/or obesity plus hypertension or obesity plus diabetes plus hypertension. * $p < 0.05$; ** $p < 0.001$; n = 784,479.

## 4. Discussion

Based on data of more than 780,000 Brazilian adults collected over a period of 16 years, it was possible to analyze the temporal trend of multimorbidity of relevant NCDs. An intense increase in the prevalence of obesity was observed in the entire population, as well as in the co-occurrence of obesity and diabetes and/or hypertension between 2006 and 2021, and especially in the most recent years (2017–2021). This increase was observed for almost all sociodemographic strata. Although obesity and co-occurrence with diabetes were higher among women, when analyzing the multimorbidity associated with the hypertension, men had a greater increase. It is noteworthy that the highest increases in multimorbidity were more related to older age groups, especially after 45 years of age and among adults with less schooling. There was also an intense increase among adults with poor self-rated health in the period.

Considering the increasing prevalence of NCDs worldwide, and the fact that many of them share common risk factors [1], it is expected that the prevalence of multimorbidity of NCDs will continue to rise. According to a recent systematic review with meta-analysis [19] with a sample of over 1.2 million adults from low- and middle-income countries, the overall prevalence of multimorbidity was 36.4%, including mainly cardiometabolic and cardiorespiratory conditions. The review presents data from different countries around the world, including Brazil, ranging from 39.3% in upper-middle-income countries to 29.2% in lower-middle-income countries. For Latin America and the Caribbean, the prevalence was 50.4% [19].

The same study found that the highest overall prevalence (81.3%) was observed among the elderly population (≥60 years) in Southern Brazil. A cross-sectional study with data from the Brazilian National Health Survey conducted in 2013 (n = 60,202 adults), using a list of 22 morbidities, reported a multimorbidity frequency of 22.2% for two or more morbidities and 10.2% for three or more [20]. Although these studies have evaluated a wider range of morbidity, reducing the comparability with the present study, the relevance of multimorbidity in low- and middle-income countries, including Brazil, is evident, and should be considered in public health policies.

A study conducted in 2013 using data from the same health survey as in the present study, included the diagnosis of diabetes, dyslipidemia, hypertension, and obesity, and reported a multimorbidity frequency of 9.8% for two and 3.3% for three NCDs among adults (<60 years) [21]. However, until the completion of this study, the current prevalence and the temporal trend of multimorbidity due to obesity, diabetes and hypertension in the Brazilian population was unknown. The present study not only updates the prevalence of multimorbidity, but also shows that the co-occurrence of obesity and diabetes and/or hypertension underwent a significant increase from 2006 (5.5%) to 2021 (9.6%), with higher intensity in the past 5 years, indicating a worrying scenario regarding the progression of NCDs in the country.

Sociodemographic factors, such as age and education, are often associated with multimorbidity of NCDs [22–24]. In agreement with the present results—which indicated a greater increase in multimorbidity among individuals with less years of education—a meta-analysis of cross-sectional studies showed that low education increased the chance of multimorbidity by 64% [22]. A similar association was observed among older Brazilian adults [23]. It has been well established that the level of education is related to access to information and health care and can interfere with adherence to health interventions [25]. Considering that the same risk behaviors can lead to different NCDs, the non-adherence to the lifestyles changes and medical treatment for a NCDs increases the risk of multimorbidity. In addition, the occurrence of multimorbidity is higher in older individuals, owing to their greater propensity for developing NCDs [26]. Considering the population aging observed in Brazil, multimorbidity has become one of the greatest challenges for health services, due to its association with functional decline and mortality [20,27].

Besides being considered a chronic disease, obesity is an important risk factor for multimorbidity, and is associated with a loss of a potential year of full health [28]. Accord-

ing to an observational multicohort study, obesity is associated with 21 non-overlapping cardiometabolic, digestive, respiratory, neurological, musculoskeletal, and infectious diseases [29]. Additionally, the interconnection between the diseases accelerates the development of obesity-related multimorbidity, and the risk of multimorbidity is higher in people suffering obesity before 50 years of age [29]. It is widely recognized that obesity predisposes to type 2 diabetes and hypertension due to the inflammatory role of excessive adipose tissue, and their co-occurrence reduces quality of life and increases the risk of cardiovascular events [30].

Considering the impact of obesity and other NCDs on health-related quality of life [31], the self-rated of the health status becomes an important indicator concerning the presence of multimorbidity. Poor self-rated health is considered a predictor of morbidity and mortality [32]. Individuals with multimorbidity experience lower well-being and quality of life, and poor self-rated health status may indicate the presence of health complications and the inadequate management of NCDs [19,32,33]. In addition, self-reported sleep problems [34], as well as other risk factors [35], are related to multimorbidity and affect the quality of life of these people [36]. In this context, the effectiveness of health interventions can benefit from the joint confrontation of chronic diseases, such as type 2 diabetes, hypertension and obesity.

The management and prevention of multimorbidity through health interventions should be offered to individuals by the primary health care. However, the heterogeneity of previous studies leads to a lack of robust evidence on effective interventions [37]. In Brazil, based on the "Global Action Plan for prevention and control of noncommunicable diseases—2013–2020" [38], the "Strategic Action Plan for Tackle Chronic Diseases and Noncommunicable Diseases in Brazil 2021–2030" [39] establishes goals to stop the growth of obesity and risk behaviors for NCDs, and reduce premature mortality from NCDs by 2% per year until 2030. Despite not specifically addressing the management of multimorbidity, actions guided by such plan, through intersectoral strategies and universal health care, can aid to control the evolution of the multimorbidity observed in the present study.

Our study presents some limitations that must be addressed. Vigitel is a survey conducted using telephone interviews; therefore, weight was self-reported and thus prone to measurement error. Such bias is especially relevant among individuals with obesity, who tend to underreport their body weight [40]. Even though the use of this self-reported information may result in measurement error, it is plausible to assume that this limitation is constant over the years, meaning that the identified trends, or even the annual variation, will not be strongly affected. Additionally, the Vigitel sample is composed only of individuals who have a landline telephone in a Brazilian capital, representing potential bias in the representativeness of the sample by reducing the participation of some population groups. It is important to highlight, however, that Vigitel uses adequate weighting factors to adjust the estimates and correct the differences between the population with and without a landline telephone, allowing the extrapolation of the results to the total Brazilian population [17]. Another limitation is related to the number of NCD included by Vigitel. The survey has continuously only analyzed obesity, hypertension and diabetes. This makes it impossible to carry out a complete study of all the multimorbidities present in the population. However, these are the most common NCDs in the country, constituting an important indicator of the health of the population. It should also be noted that, despite the multifactorial nature of NCDs, obesity represents an important risk factor for these diseases. Although the study was directed towards the concomitant presence of obesity with diabetes and hypertension, according to information available on Vigitel, other NCDs are directly associated with the presence of obesity [41]. In this sense, studies with a greater number of comorbidities associated with obesity are still needed to better understand this scenario.

Despite these limitations, our study stands out as the first to investigate the prevalence of NCD multimorbidity among Brazilian adults over a long period of time (16 years). The evaluation of the temporal trend of the co-occurrence of the studied NCDs makes it possible

to better understand the multimorbidity trajectory over the years and management policies and thus, support actions for the prevention and management of multimorbidity. These results reinforce the need to intensify actions to promote health and protect the population from modifiable risk behaviors associated with NCDs, especially those capable of leading to greater multimorbidity. Conducting this investigation through populational surveys is essential to expanding the knowledge of the improvements required in health policies.

## 5. Conclusions

Multimorbidity prevalence increased in the population for the total period and the most recent period studied. The highest increase in the period occurred especially among men, older adults, and adults with less education. Adults with multimorbidity had a poor self-rated health during the analyzed period. These results reinforce the need to expand and strengthen public health actions focused on individuals with multimorbidity especially with obesity.

**Author Contributions:** Conceptualization, T.C.M.C. and R.M.C.; Methodology, T.C.M.C. and R.M.C.; Formal Analysis, T.C.M.C. and M.M.S.; Writing—Original Draft Preparation, T.C.M.C., M.M.S., T.M.S., I.P.A.V. and L.E.S.S.; Writing—Review and Editing, all authors. All authors have read and agreed to the published version of the manuscript.

**Funding:** The work was supported by the Coordenação de Aperfeiçoamento de Pessoal de Nível Superior (CAPES) [grant number 001 (Scholarship for M.M.S.)] and the Conselho Nacional de Desenvolvimento Científico e Tecnológico (grant number 311170/2019-6 (Scholarship for R.M.C.)).

**Institutional Review Board Statement:** Vigitel's was authorized by the Brazilian Committee on Ethics in Research with Human Beings of the Ministry of Health (65610017.1.0000.0008).

**Informed Consent Statement:** Oral informed consent was obtained from all subjects involved in the study during the interview.

**Data Availability Statement:** Vigitel's data are available on the official Ministry of Health website: (http://svs.aids.gov.br/download/Vigitel/ (accessed on 3 February 2023)).

**Conflicts of Interest:** The authors declare no conflict of interest.

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
