# Peer review of "Temporal Trend of Multimorbidity of Noncommunicable Diseases among Brazilian Adults, 2006–2021"

_2673-4168, doi:10.3390/obesities3010007_

Round 1

Reviewer 1 Report

The authors are intersting in "identify the temporal trend of multimorbidity of noncommunicable 11 disease (NCDs) among Brazilian adults over a 16-year period-of-time."

Disadvantage of this study:

1. Although the study seems complete, unfortunately it does not make a comparison with similar studies from other countries.

2. This study is limited to the population of Brazil only.

Author Response

Response to Reviewer 1 Comments

We are grateful for the revision carried out by the reviewers. All changes and their justifications were noted in the letters of each reviewer.

Reviewer 1: Although the study seems complete, unfortunately it does not make a comparison with similar studies from other countries.

Authors: Thank you for the suggestion. We used a systematic review to compare our results with other countries. The systematic review used data from more than 1.2 million adults in different countries around the world. We entered data from other countries based on the results of this systematic review with upper middle and lower middle income countries, including Latin America.

“The review presents data from different countries around the world, including Brazil, ranging from 39.3% in upper-middle-income countries to 29.2% in low-er-middle-income countries. For Latin America and the Caribbean, the prevalence was 50.4% [19].”

Reviewer 1: This study is limited to the population of Brazil only.

Authors: Our study was based on the largest and longest health survey in Brazil. We believe that these data can expand knowledge about multimorbidity’s in high-middle-income countries, such as Brazil.

Reviewer 2 Report

This paper is well organized and written. However, it has one giant shortcoming. Though claiming to address multi-morbidity of non-communicable diseases, it only considers obesity, diabetes and hypertension. Each of these diseases is also multi-morbid with numerous other diseases and the onset of each is also influenced by all pre-existing conditions, chronic environmental exposures, genetics, lifestyle and oxidative stress the individuals. These other factors are not addressed. Accordingly, the paper is incomplete. The addition of these other factors to it will make this a spectacular paper.

Author Response

(The authors gave the same response as above.)

Round 2

Reviewer 2 Report

The authors do not properly address the effects of diseases and conditions other than obesity, diabetes and hypertension. It is well known that numerous other pre-existing conditions also are influenced by obesity and that obesity can serve as a pre-existing condition that triggers other non-communicable diseases. The authors briefly acknowledge this, yet fail to take such effects into consideration. 

The paper can be published in its amended form, but should contain a limitation statement regarding causes and effects of obesity.

Author Response

Response to Reviewer 2 Comments 

The change and its justifications were noted in the letter. 

Reviewer 2:  The authors do not properly address the effects of diseases and conditions other than obesity, diabetes and hypertension. It is well known that numerous other pre-existing conditions also are influenced by obesity and that obesity can serve as a pre-existing condition that triggers other non-communicable diseases. The authors briefly acknowledge this,  yet fail to take such effects into consideration.  

The paper can be published in its amended form, but should contain a limitation statement regarding causes and effects of obesity. 

Authors: Thanks for your suggestion. We understand your question. The limitation of the study was inserted. 

“It should also be noted that, despite the multifactorial nature of NCDs, obesity represents an important risk factor for these diseases. Although the study was directed towards the concomitant presence of obesity with diabetes and hypertension, according to information available on Vigitel, other NCDs are directly associated with the presence of obesity [41]. In this sense, studies with a greater number of comorbidities associated with obesity are still needed to better understand this scenario.” 
